# Virus transmission via honey bee prey and potential impact on cocoon-building in labyrinth spiders (*Agelena labyrinthica*)

**Daniel Schläppi**[1]*, **Nor Chejanovsky**[2,3], **Orlando Yañez**[2], **Peter Neumann**[2]

**1** School of Biological Sciences, University of Bristol, Bristol, United Kingdom, **2** Institute of Bee Health, Vetsuisse Faculty, University of Bern, Bern, Switzerland, **3** Department of Entomology, Agricultural Research Organization, The Volcani Institute, Rishon LeZion, Israel

* schlaeppi@mail.ch

**Data Availability Statement:** All data is available from figshare at doi:10.6084/m9.figshare. 21252489. (https://doi.org/10.6084/m9.figshare. 21252489.v1).

## Abstract

Interspecies transmission of RNA viruses is a major concern for human and animal health. However, host-range, transmission routes and especially the possible impact of these viruses on alternative hosts are often poorly understood. Here, we investigated the role of the labyrinth spider, *Agelena labyrinthica*, as a potential alternative host of viruses commonly known from western honey bees, *Apis mellifera*. Field-collected spiders were screened for Acute bee paralysis virus (ABPV), Black queen cell virus, Chronic bee paralysis virus, Deformed wing virus type A and B (DWV-B), Israeli acute paralysis virus, Lake Sinai virus and Sacbrood virus. In a laboratory experiment, labyrinth spiders were fed with ABPV and DWV-B infected honey bees or virus free control food. Our results show that natural infections of *A. labyrinthica* with these viruses are common in the field, as 62.5% of the samples were positive for at least one virus, supporting their wide host range. For DWV-B, the laboratory data indicate that foodborne transmission occurs and that high virus titres may reduce cocoon building, which would be the first report of clinical symptoms of DWV in Araneae. Since cocoons are tokens of fitness, virus transmission from honey bees might affect spider populations, which would constitute a concern for nature conservation.

## Introduction

Increasing evidence suggests alarming declines both in abundance and diversity of terrestrial arthropods [1–4]. As arthropods provide crucial ecosystem services, including pollination and natural pest control, the observed trends could have far-reaching consequences [5, 6]. Pathogens causing emerging infectious diseases pose a substantial threat to both human- and animal health [7, 8], and they have been identified as one of the drivers of the observed declines [9].

Many pathogens, particularly RNA viruses, are known to be multi-host species [10]. Properties of RNA viruses such as error prone replication and large population sizes enable elevated rates of adaptive evolution, facilitating the crossing of species barriers [11, 12]. Accordingly, there is increasing consensus for virus transmission between managed western honey bees (*Apis mellifera*) and wild bees, possibly contributing to recent pollinator declines [13]. Several

**Funding:** This research was funded by the Béatrice Ederer-Weber Foundation (P.N.; https://www.ederer-stiftung.ch/startseite), the Vinetum Foundation (P.N.; https://www.vinetum.ch/de/Home), the Stiftung Dreiklang für ökologische Forschung und Bildung (D.S., P.N.) and the Swiss National Science Foundation (D.S., P500PB_206883 / 1; https://www.snf.ch/en). Neither the design of the study, data collection, analyses and interpretation, writing of the manuscript nor the decision to publish the results were influenced by the funders.

**Competing interests:** The authors have declared that no competing interests exist.

viruses first described in and commonly associated with *A. mellifera*, often referred to as honey bee viruses, have been detected across the wider arthropod community [14–17]. Furthermore, foodborne transmission, i.e. consumption of infected prey, has been suggested to enable the infection of predaceous or scavenging arthropods, such as ants and wasps [18–22]. However, there is a dearth of data on the prevalence of these viruses in the class Arachnida. Presence of honey bee associated viruses has been confirmed in the ectoparasitic mites, *Varroa destructor* and *Tropilaelaps mercedesae*, both well known honey bee parasites [23, 24]. The mites carry and efficiently vector viruses, but if they are actual biological hosts remained unclear until recently. While evidence suggests non-propagative vectoring of Deformed wing virus (DWV) type A, DWV-B appears to replicate in *V. destructor* [25, 26].

Apart from mites, we are aware of only two studies reporting the presence of Black queen cell virus, Deformed wing virus (DWV), Kashmir bee virus (KBV), Moku virus (MKV) and Sacbrood virus (SBV) in a very limited number of Arachnid samples [14, 27]. Two spider samples were positive for the minus-sense strand of DWV as well, suggesting potential replication in these hosts [14]. However, it is unclear if these might be false positives due to viral particles consumed in conjunction with infected prey [20]. Consequently, the role of spiders as potential alternative hosts remains poorly understood. As generalist predators, spiders play crucial roles in food webs and have been recognized as important biological control agents [28]. Furthermore, despite numerous studies detecting honey bee associated viruses in a large number of alternative hosts (e.g. more than 70 species for DWV) [29], only few studies have actually addressed possible impacts of these viruses on their novel hosts, i.e. clinical symptoms. Pathogenicity has been studied mostly in *Bombus* spp. (for an overview see [30]). Outside of bees, there are only a few studies reporting clinical symptoms in wasps (*Vespa crabro*) and ants (*Lasius niger*), respectively [22, 31].

Here, we investigate the role of spiders as potential alternative hosts of viruses initially described from *A. mellifera*, using the labyrinth spider, *Agelena labyrinthica*, commonly found in Europe [32]. Labyrinth spiders frequently consume honey bee prey and are therefore in principle exposed to foodborne virus transmission (Fig 1). We screened field-collected samples of these spiders for viruses commonly detected in honey bees, i.e. Acute bee paralysis virus

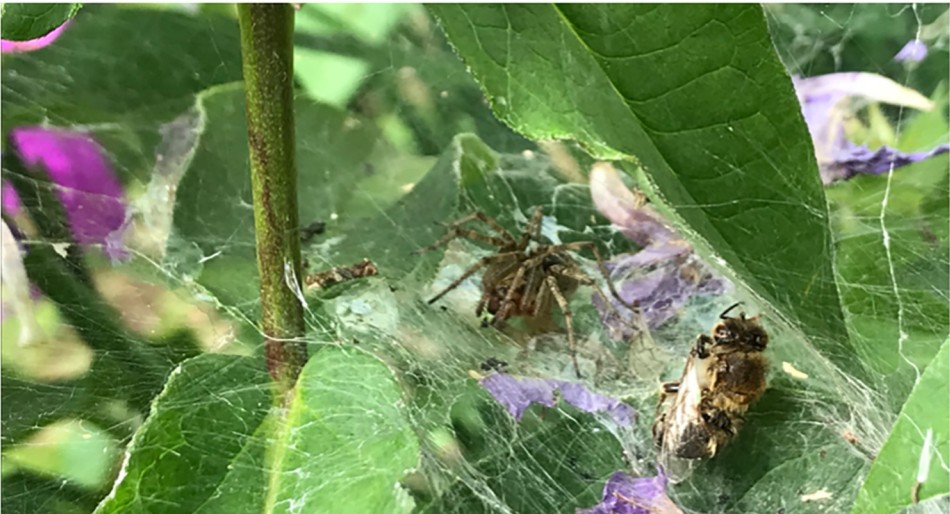

**Fig 1. Labyrinth spider (*Agelena labyrinthica*) sitting in her web in the field with a captured honey bee prey (*Apis mellifera*).**

(ABPV), Black queen cell virus (BQCV), Chronic bee paralysis virus (CBPV), DWV-A, DWV-B, Israeli acute paralysis virus (IAPV) and SBV). In addition, we empirically tested whether foodborne transmission enables ABPV and DWV-B to infect *A. labyrinthica* using a laboratory assay. Both DWV-B and ABPV are positive-sense RNA viruses and harmful honey bee pathogens with a known wide host range [29, 30, 33]. The opisthosoma contains the majority of the digestive system, including the extensive diverticula of the midgut where most of digestion takes place [34]. Thus, to reduce chances of false positive results due to virus particles in the intestine, the prosoma and the opisthosoma were tested separately for each spider of the laboratory assay and only the prosoma was analysed from samples of the field screening.

## Materials and methods

### Sampling of spiders & experimental set-up

*Agelena labyrinthica* (*N* = 40) were sampled around Bern, Switzerland (July 2016: *N* = 30, 46˚56'15.2"N 7˚22'41.9"E; July 2017: *N* = 10, 46˚55'52.7"N 7˚25'58.8"E) and taxonomically identified using morphometrics [32]. Six spiders from 2016 and all samples from 2017 were immediately frozen and stored at -80 ˚C until further processing. The remaining 24 spiders were transferred into cages (Θ = 97 mm, height = 128.5 mm). The cages were filled with 5 mm of water retaining granules and 10 mm of biological potting soil. Further, each of the cages contained a stick and a dried leaf (*Ilex aquifolium*) as starting point for the funnel webs. The cages were maintained at room temperature (19–23 ˚C) and protected from direct sunlight. The nets and the soil were sprayed regularly with water. Until the start of the feeding experiment, the spiders were fed weekly with fruit flies (*Drosophila hidey*). The spiders were randomly assigned to two treatments, differing in their feeding regime at two experimental feeding events (12.08.2016, 22.08.2016; see S1 Fig for a timeline). In each feeding session, controls (*N* = 5, all ♀) were fed with one cricket each (*Acheta domesticus*), and spiders in the treatment group (N = 19, ♀ = 16, ♂ = 3) received one freshly emerged honey bee (*A. mellifera*) artificially infected with both ABPV and DWV-B. The ideal food for the controls would have been non infected honey bees. However, low level latent virus infections are highly likely in honey bees [35], and therefore crickets were used as a secure option. Spiders lay their eggs in protective silk cases, often referred to as cocoons or egg sacs [32]. Thus, cocoon building was recorded for all female spiders as a proxy for reproductive output, aka token of fitness.

Two spiders (both ♂) were found dead in their nest (17.08.2016, 22.08.2016) and frozen at -80˚C until further processing. Four days after the last feeding event (26.08.2016), all remaining spiders were frozen at -80˚C. No ethical approval or permit was required to sample and work with the invertebrate species used in this study and all experiments were performed in accordance with relevant guidelines and regulations of local authorities.

### Preparation of virus-infected honey bees

To ensure high virus loads, ABPV and DWV-B were propagated in honey bee pupae (*A. mellifera*) using standard methods [36]. Red to dark-eyed pupae were collected from sealed worker brood frames of two local *A. mellifera* colonies (08.08.2016). Then, the pupae were microinjected laterally between the second and third segment of the abdomen with two μL of a solution containing both ABPV and DWV-B. Injected pupae were incubated at 34.5˚C, ≥50% relative humidity and darkness until emergence [37]. The emerging adults were fed to the spiders from the treatment group while controls received crickets (12.08.2016; Fig 2). Two weeks later, the same steps were repeated with pupae sampled at the 18.08.2016 and feeding at the 22.08.2016.

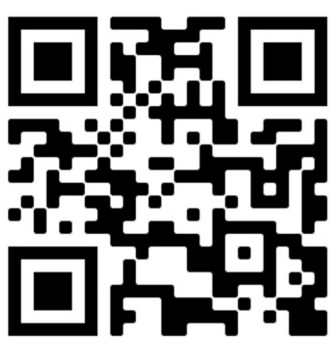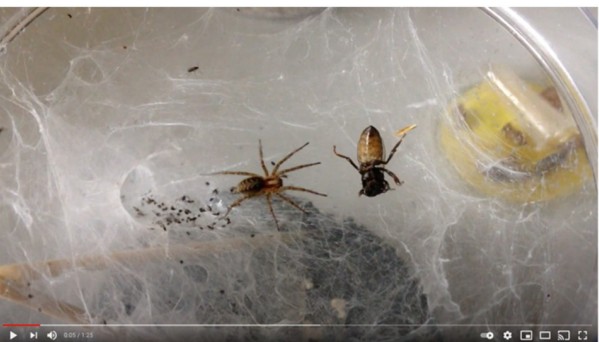

**Fig 2. QR code to access a video showing a labyrinth spider (Agelena labyrinthica) capturing an adult honey bee (Apis mellifera) in the feeding experiment (URL: https://youtu.be/kO5B2Z7OiNw).**

## Virus analyses

The prosoma of all spiders from the field screening ($N$ = 16) and the prosoma as well as the opisthosoma of all spiders from the feeding experiment ($N$ = 24) were analysed for viruses. Spiders and crickets ($N$ = 16) from the feeding experiment were tested for ABPV and DWV-B. The remaining samples were screened additionally for BQCV, CBPV, DWV-A, IAPV, LSV and SBV. For the RNA extraction, samples were crushed in 2 ml Eppendorf® tubes containing a 3 mm metal bead and filled with 100 µl TN buffer (100 mM Tris, 100 mM NaCl, pH 7.6) using a Retsch® MM 300 mixer mill for 1 min at the frequency 25 1/s [38]. Fifty microliters of the homogenate were used for the RNA extraction following the manufacturer's recommendation of a NucleoSpin® RNA II kit (Macherey-Nagel, Oensingen, Switzerland). Thirty microliters were used to elute the RNA, which was then stored at −80 ˚C until further processing [38]. At the first extraction step 200µg/ml Ambion™ RNA Control 250 (RNA250) was added as exogenous internal reference standard to monitor the efficiency of the RNA purification and cDNA synthesis [39].

An M-MLV RT Kit (Promega, Dübendorf, Switzerland) was used for the reverse transcription to obtain cDNA following the manufacturer's recommendations. Random hexamer oligonucleotide (0.75 µL; 100 µM), template RNA and $H_2O$ (final reaction volume 17.75 µL) were incubated in a Thermocycler (Biometra, Analytik Jena, Jena, Germany). Then, 5×Buffer (5 µL), nucleoside triphosphate (dNTP; 1.125 µL; 10mM) and reverse transcriptase (M-MLV; 1 µL) were incubated for 60 min at 37 ˚C. After synthesis, the resulting cDNA was diluted 1/5 and stored at −25 ˚C.

To estimate viral titres, real-time reverse transcription-quantitative polymerase chain reaction (RT-qPCR) using a KAPA SYBR® FAST Universal qPCR kit (KAPA Biosystems, Wilmington, North Carolina, United States). On each plate samples were run in duplicate for both the exogenous internal reference and the viruses of interest using 3 µL of diluted cDNA, 6 µL KAPA SYBR® green reaction mix, 2.52 µL $H_2O$ and 0.24 µL from each of a specific primer pair (Table 1) in a final reaction volume of 12 µL [36]. Additionally, each plate contained a ten-fold serial dilution of purified PCR products acting as standard curves for the virus of interest and the exogenous internal reference and two no-template negatives [40]. The reaction was processed with an ECO™ Real-Time PCR machine (Illumina, San Diego, California, United States). The qPCR cycling profile was set as follows: 40 cycles of denaturation (3 s, 95˚C) plus annealing, extension and data collection (30 s, 57˚C), preceded by a period of incubation (3 min, 95 ˚C). The specificity of the PCR products was verified using a melting curve analysis (reading the fluorescence at 0.5 ˚C intervals between 55 and 95 ˚C) following the amplification.

**Table 1. Primers used for the relative virus quantification of viruses in spiders, *Agelena labyrinthica*.** Primers used for sequencing of PCR products are marked with asterisk (*).

| Family | Target | Primer | Sequence (5′–3′) | [bp] | Ref |
|---|---|---|---|---|---|
| Dicistroviridae | ABPV* | ABPV F6548 | TCATACCTGCCGATCAAG | 197 | [43] |
| | | ABPV B6707 | CTGAATAATACTGTGCGTATC | | |
| | IAPV | IAPV-F6627 | CCATGCCTGGCGATTCAC | 203 | [43] |
| | | KIABPV-B6707 | CTGAATAATACTGTGCGTATC | | |
| | BQCV | BQCV-qF7893 | AGTGGCGGAGATGTATGC | 294 | [43] |
| | | BQCV-qB8150 | GGAGGTGAAGTGGCTATATC | | |
| Iflaviridae | DWV-A | DWV F8668 | TTCATTAAAGCCACCTGGAACATC | 136 | [23] |
| | | DWV B8757 | TTTCCTCATTAACTGTGTCGTTGA | | |
| | DWV-B | VDV F2 | TATCTTCATTAAAACCGCCAGGCT | 139 | [44] |
| | | VDV R2a | CTTCCTCATTAACTGAGTTGTTGTC | | |
| | DWV-B* | VDV F1409 | GCCCTGTTCAAGAACATG | 413 | [43] |
| | | DWV B1806 | CTTTTCTAATTCAACTTCACC | | |
| | SBV | SBV-qF3164 | TTGGAACTACGCATTCTCTG | 335 | [43] |
| | | SBV-qB3461 | GCTCTAACCTCGCATCAAC | | |
| Non assigned | CBPV | CBPV1-qF1818 | CAACCTGCCTCAACACAG | 296 | [43] |
| | | CBPV1-qB2077 | AATCTGGCAAGGTTGACTGG | | |
| | LSV-U | qLSVU-F-2350 | TTATCTCGCGCCGCCACCTC | 188 | [45] |
| | | qLSVU-R-2538 | AGAGGGTACCGCGACACCCATG | | |
| Non applicable | RNA250 | RNA 250-F | TGGTGCCTGGGCGGTAAAG | 227 | [39] |
| | | RNA 250-R | TGCGGGGACTCACTGGCTG | | |

Based on the experimental dilution factors and the q-PCR output, including the data of the standard curves, the estimated numbers of viral copies per sample, i.e. virus titres, were derived. Samples were considered negative if no peak or a shifted peak was observed in the melting curve analysis, or if the Cq value exceeded the one of the respective $H_2O$ negative controls. For both ABPV and DWV-B, a titre detection threshold, defined as the maximum of value of hypothetical titres calculated from Cq values of negative samples [41], was subtracted from calculated titres and negative samples were assigned zero viral copies. Then, titres were log-transformed to account for their exponential nature. Throughout the manuscript logarithmic values of titres (Log10 genomic copies/sample) are reported. To avoid logarithm of zero, 1 was added to each titre prior to transformation. Further, for both body parts, infections with either virus were categorized into low- and high-level infections (HLI), based on titres being above or below $10^7$ genomic viral copies, as titres above this threshold correlate with clinical symptoms and consequently overt infections in honey bees [42]. Samples with high concentration of the target viruses were selected for further confirmation of their identity. Purified PCR products from each viral target were sequenced twice (one time from each 5Samples wextreme). The overlapped sequences were uploaded to GenBank: ABPV—OQ272302; DWV-B—OQ272303).

## Statistical analyses

All analyses were performed using R version 3.5.1 [46]. Data and model residuals were checked for normal distribution with the Shapiro-Wilk test and homogeneity of variances between groups using the Levene's test to select appropriate statistical analyses. To compare virus titres between the treatments, Mann–Whitney *U* tests were applied. A one-tailed N-1 Pearson's Chi-Square test was applied to test whether positive samples and HLI's were commoner among

**Table 2. Detection of viruses known from honey bees (*Apis mellifera*) in field-collected labyrinth spiders (*Agelena Labyrinthica*; N = 16).** Samples tested positive are marked with [+] and negative ones with [−]. The number of samples indicates how many individually analysed spider samples matched each virus profile.

| Number of samples | Virus | | | | | | | |
|---|---|---|---|---|---|---|---|---|
| | ABPV | BQCV | CBPV | DWV-A | DWV-B | IAPV | LSV-U | SBV |
| 3 | + | - | - | + | + | - | - | + |
| 1 | + | - | - | - | + | - | - | - |
| 1 | - | - | - | + | - | - | - | - |
| 5 | - | - | - | - | + | - | - | - |
| 6 | - | - | - | - | - | - | - | - |

virus treated spiders [47]. Per definition only female spiders were included to test for differences in the proportion of spiders building a cocoon using a N-1 Pearson's Chi-Square. For the comparison of ABPV titres, DWV titres and overall virus titres (sum of DWV and ABPV) between females that have built a cocoon and others that did not, linear mixed effect modelling was used with titre as response variable, presence or absence of a cocoon as explanatory variable and treatment as random factor to account for different food sources provided in the two groups (crickets or infected honey bees). A Paired Samples Wilcoxon test was applied to compare virus titres between pro- and opisthosoma. To test for the correlation of titres between the two body parts a Kendall rank correlation test was used.

## Results

Of the surveyed viruses (ABPV, BQCV, CBPV, DWV-A, DWV-B, IAPV, LSV, SBV) only ABPV, DWV-A, DWV-B and SBV were detected in field collected samples of *A. labyrinthica* (Table 2). In 62.5% of the tested spiders at least one virus was detected. DWV-B was the most common with more than half of the samples being positive (56.25%), followed by ABPV (25%), DWV-A (25%) and SBV (18.75%). In 40% of the spiders, presence of more than one virus was found.

A majority of spiders (83.33%) from the feeding experiment, including controls, were positive for ABPV and DWV-B (Table 3) while the crickets tested negative for both viruses. Sequencing of the PCR products confirmed the identity of both ABPV and DWV-B (GenBank accession: ABPV, MT141130.1, 100% identity, 100% query cover, DWV-B, MN538209.1, 99.05% identity, 100% query cover). Virus titres in these spiders (prosoma and opisthosoma combined) ranged from 4.23 to 10.4 log genomic viral copies per sample for ABPV and from 3.34 to 9.61 for DWV-B. For both viruses, titres of the virus treatment were higher than in the controls, but the differences were not significant (ABPV: W = 26, $p$ = 0.14; DWV-B: W = 21,

**Table 3. Titres of ABPV and DWV-B reported as medians [1st 3rd quartiles] in labyrinth spiders (*Agelena Labyrinthica*; N = 24) after an experimental feeding assay with infected honey bee pupae (*Apis mellifera*).**

| Virus | Treatment | Prosoma | | | Opisthosoma | | |
|---|---|---|---|---|---|---|---|
| | | Pos[1] | Titre | HLI[2] | Pos | Titre | HLI |
| ABPV | Control (N = 5) | 5 (100%) | 4.60 [4.37; 5.16] | 1 | 4 (80%) | 5.08 [4.64; 5.41] | 0 |
| | Virus (N = 19) | 18 (94.7%) | 5.33 [4.67; 6.24] | 4 | 16 (84.2%) | 6.15 [5.25; 7.75] | 5 |
| DWV-B | Control (N = 5) | 3 (60%) | 4.45 [3.90; 5.45] | 0 | 1 (20%) | 4.57 [4.57; 4.57] | 0 |
| | Virus (N = 19) | 16 (84.2%) | 5.07, [4.51; 7.17] | 5 | 13 (63.2%) | 7.58 [5.56; 8.19] | 7 |

[1] Pos–Number (percentage) of samples tested positive

[2] HLI–High-level infections: Number of samples with virus titres $<10^7$ genomic viral copies

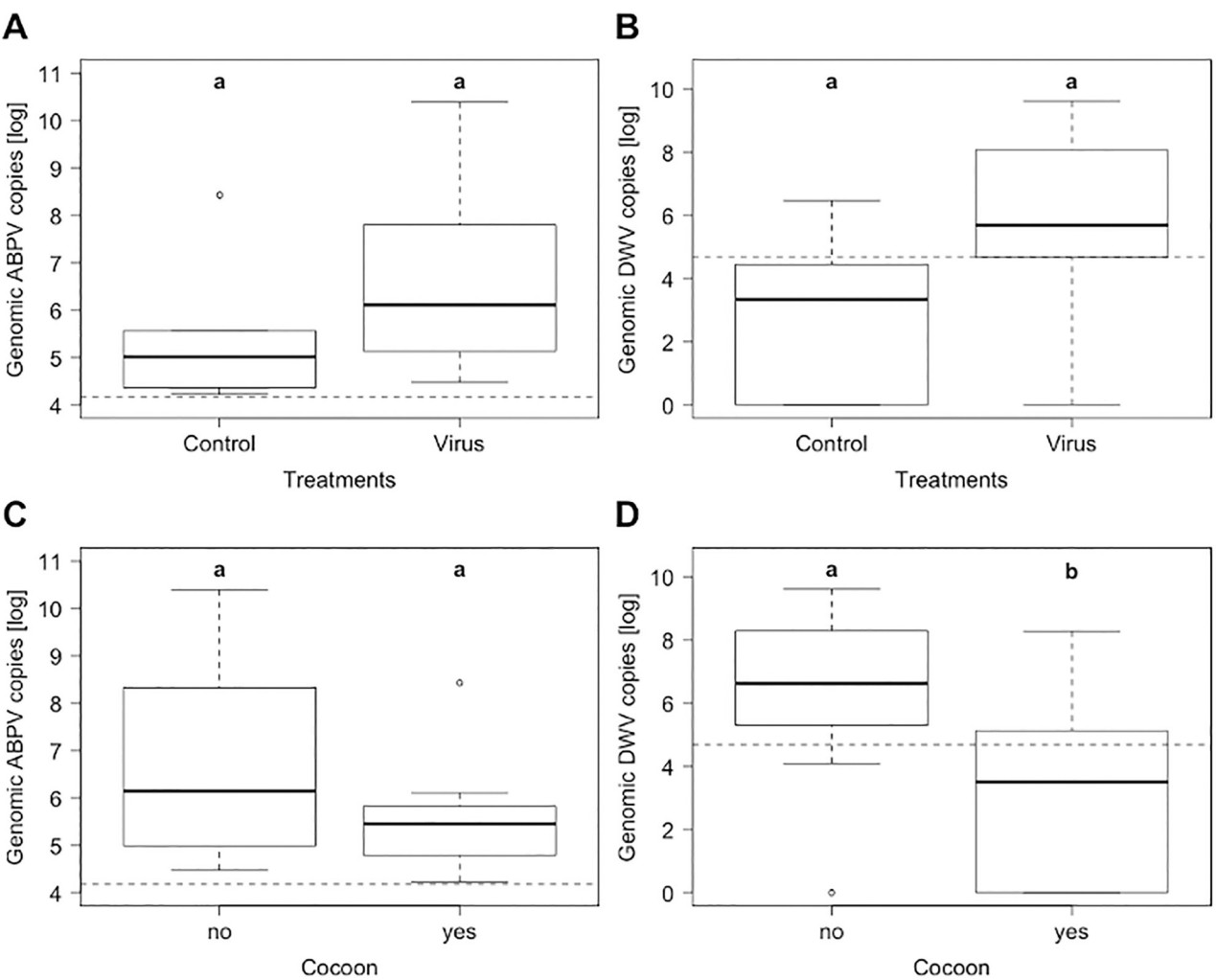

**Fig 3.** Titres of (**A,C**) Acute bee paralysis virus (ABPV) and (**B,D**) Deformed wing virus type B (DWV-B) in labyrinth spiders (*Agelena labyrinthica*) from the experimental feeding trial: (**A & B**) Virus titres for the two treatment groups that differed with regard to the feeding regime (controls = no virus, treatment = virus); (**C & D**) female spiders that either did or did not build a cocoon. Boxplots are displayed with the inter-quartile-ranges (box), medians (black line in box) and outliers (dots). Dashed lines represent virus detection thresholds. Significant differences ($p < 0.05$) between the groups are indicated by small bold letters (**a**, **b**).

$p = 0.064$; Fig 3A & 3B). Samples of the virus treatment were not significantly more often positive for DWV-B than controls ($X^2_{(1)} = 2.37$, $p = 0.062$). No difference was found for ABPV, as all samples were positive. Nonetheless, HLI's with at least one of the two viruses were detected more often in spiders of the virus treatment (chi-square test: $X^2_{(2)} = 2.85$, $p = 0.046$). Apart from one control sample with high levels of ABPV in the prosoma, HLI's were only detected in virus treatments.

Further, several females built a cocoon over the duration of the experiment (42 days). In the controls, the proportion of cocoon building spiders (5 out of 5) was significantly higher compared to the virus treatment (4 out of 16; $X^2_{(1)} = 8.33$, $p = 0.004$). Furthermore, cocoon-building spiders had significantly lower overall virus titres (sum of DWV and ABPV; $X^2_{(1)} = 8.01$, $p = 0.005$) and DWV titres ($X^2_{(1)} = 6.2$, $p = 0.013$; Fig 3D) in comparison to spiders without cocoons irrespective of treatment group. The same trend was observed for ABPV titres, but the difference was not significant ($X^2_{(1)} = 2.49$, $p = 0.11$, Fig 3C).

For neither virus a significant difference in virus titres was found between pro- and opistho-soma (ABPV: $z = -0.18$, $p = 0.86$; DWV-B: $z = -0.69$, $p = 0.49$). All prosomas with a HLI of DWV-B were associated with opisthosomas, which were highly infected as well. Two additional opisthosomas with HLI were detected, with their corresponding prosomas having low infection levels. For ABPV, HLI's matched only in two cases between the pro- and the opisthosoma. Each three prosomas and opisthosomas had HLI's while their counterpart did not. Nonetheless, there was an overall correlation of the titres between the two body parts (ABPV: $R = 0.38$, $z = 2.59$, $p = 0.01$; DWV-B: $R = 0.52$, $z = 2.59$, $p < 0.001$).

## Discussion

To our knowledge, this is the first study investigating transmission of honey bee-associated viruses to spiders. Our data indicate that virus transmission from honey bees to *A. labyrinthica* can be common in the field and that foodborne transmission, at least for DWV-B, is an underlying mechanism. Furthermore, the experimental results suggest that high DWV-B titres may impair cocoon building, therewith likely reducing spider fitness. This is the first report of clinical symptoms of honey bee associated viruses in the order Araneae and constitutes a concern for nature conservation.

Screening of field-collected labyrinth spiders revealed the presence of ABPV, DWV-A, DWV-B and SBV. Thereby, we expand the knowledge on these viruses, which still is very limited for the order Araneae [14, 17, 29]. Unsurprisingly, DWV-B was detected the most, matching the prevalence of this virus in honey bees [29, 48]. These findings are in line with the wide host-range of DWV [29], and one more species, the first from the order Araneae, can be added to the range of susceptible hosts. However, it is unclear these are true host shifts, i.e. viruses invading and establishing in a new host species [49], or whether the viruses are simply generalist arthropod viruses. Even though most of these viruses were described first in honey bees, it remains uncertain if bees were the original host.

Viruses, including high-level infections analogous to overt infections in honey bees, suggesting viral replication, were detected both in the prosoma and the opisthosoma of spiders. As is always the case for predatory arthropods, virus particles taken up with recently consumed food items can lead to false positives and bias the results [19]. Although gut contents may have been cleared out prior to testing, as spiders defaecate quite regularly [50], we cannot exclude this experimental artifact. This also holds true for the prosoma because a fraction of the midgut expands as caeca into the cephalothorax [34]. Usually, replication of the virus is confirmed by the presence of negative-sense strand RNA, a token of virus replication. Because the recently consumed honey bee prey is full of active replicating virus, we argue that for our experimental design the detection of negative-sense strand RNA would not constitute conclusive evidence. However, if virus loads would simply reflect particles recently taken up with infected prey, we would expect to find most of the virus in the ophistosoma, as it holds a majority of the digestive system [34], whereas our data show that there are no differences in virus titres between the pro- and ophistosoma. Consequently, the presence of virus particles in the gut is not sufficient to explain the observed patterns, thus indicating potential viral replication.

In the experimental assay, the proportion of samples tested positive for ABPV or DWV B was not significantly different between treatments and controls. Given the results of the field screening, it is not surprising to confirm virus presence in control samples that were not experimentally exposed. It is likely that these viruses are commonly present as latent infections, potentially from earlier contacts with a contaminated source. However, HLI's were significantly more common in spiders of the treatment group indicating that foodborne transmission, i.e. the consumption of infected prey, is indeed an underlying mechanism enabling transmission

of viruses from honey bees to spiders. Since all controls were positive for ABPV, the laboratory feeding data for the transmission of this virus remain inconclusive. Together with previous studies suggesting that foodborne transmission is a common exposure route for arthropod predators or scavengers [18–22, 51], these findings highlight the importance of this pathway for interspecies virus transmission. The laboratory feeding clearly reflects a worst-case scenario with high virus loads. Nonetheless, positive samples from the field suggest that virus transmission to spiders frequently occurs under natural conditions. Given the abundance of managed western honey bees, the prevalence of these viruses in this host species and the frequency at which honey bee prey can be detected in webs of labyrinth spiders, they are the most likely source. However, in light of the wide host range of these viruses, it is well possible that the spiders obtain them from other non-honey bee sources, i.e. any other arthropod prey or via any other exposure route. As we are lacking data on viral prevalence in the surrounding apiaries or the local arthropod community, we can of course not draw any conclusion on the actual transmission dynamics. Whether there are other ways for spiders to become infected with these viruses, e.g. consumption of contaminated pollen, is poorly understood [21, 52]. Furthermore, if there are pathways for the viruses to be transmitted back to honey bees remains unknown.

Interestingly, elevated viral loads coincide with a reduction in cocoon building, suggesting that this constitutes a clinical symptom of virus infection in spiders with likely effects on fitness. Spiders that did not build a cocoon had elevated titres of ABPV and significantly higher titres of DWV-B compared to spiders that did not; irrespective of treatment, i.e. food source. Immune defence and reproduction are both physiologically and energetically demanding processes frequently found to trade off [53]. Our findings might reflect a similar scenario in which the activation of the immune system in response to an infection is reducing reproductive output, i.e. preventing cocoon building. Clinical symptoms are seen as a clear sign of overt infections with ongoing viral replication [54], which could explain our observations and would indicate that spiders are indeed true hosts. However, the sample size in this study is rather limited and further evidence is required to confirm active replication in these potential hosts, e.g. using fluorescence- *in situ* -hybridization [26]. Nonetheless, our findings suggest that *A. labyrinthica*, and possibly other spiders are potential hosts of viruses commonly associated with honey bees.

## Conclusions

Our survey and experimental findings show that *A. labyrinthica* spiders regularly host honey bee associated viruses in the field, probably due to foodborne infection with potential effects on spider populations. Despite vast amounts of literature on virus transmission among the arthropod community only a few studies address potential impacts of successful infections. Spiders and other generalist predators play an important role in terrestrial ecosystems. Hence, fitness-reducing virus host shifts have the potential to disrupt ecosystem functioning and are of great concern.

## Supporting information

**S1 Fig. Experimental timeline.** Chronological order of key events starting with the collection of spiders in the field until the experiment got terminated with the freezing of all spider samples at -80˚C.
(TIF)

## Acknowledgments

We wish to thank Kaspar Roth and Elodie Christina for technical support and Laura Bosco for fruitful discussions and courageous help with the field-sampling.

## Author Contributions

**Conceptualization:** Daniel Schläppi, Peter Neumann.

**Data curation:** Daniel Schläppi.

**Formal analysis:** Daniel Schläppi.

**Funding acquisition:** Daniel Schläppi, Peter Neumann.

**Investigation:** Daniel Schläppi, Orlando Yañez.

**Methodology:** Nor Chejanovsky, Orlando Yañez.

**Project administration:** Daniel Schläppi, Peter Neumann.

**Supervision:** Nor Chejanovsky, Orlando Yañez, Peter Neumann.

**Visualization:** Daniel Schläppi.

**Writing – original draft:** Daniel Schläppi.

**Writing – review & editing:** Daniel Schläppi, Nor Chejanovsky, Orlando Yañez, Peter Neumann.

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
