## [Decision Letter · Decision Letter 0]

5 Dec 2022

PONE-D-22-27101Virus transmission via honey bee prey and potential impact on cocoon-building in labyrinth spiders (*Agelena Labyrinthica*)PLOS ONE

Dear Dr. Schlaeppi,

Thank you for submitting your manuscript to PLOS ONE. After careful consideration, we feel that it has merit but does not fully meet PLOS ONE’s publication criteria as it currently stands. Therefore, we invite you to submit a revised version of the manuscript that addresses the points raised during the review process.

We look forward to receiving your revised manuscript.

Kind regards,

Olav Rueppell

Academic Editor

PLOS ONE

Journal Requirements:

4.  Please remove your figures from within your manuscript file, leaving only the individual TIFF/EPS image files, uploaded separately. These will be automatically included in the reviewers’ PDF.

Additional Editor Comments:

Particularly reviewer #1 raised some important points, including that we cannot publish the study at all unless the raw data are publicly accessible.

Reviewers' comments:

Reviewer's Responses to Questions

**Comments to the Author**

1. Is the manuscript technically sound, and do the data support the conclusions?

Reviewer #1: Yes

Reviewer #2: Yes

2. Has the statistical analysis been performed appropriately and rigorously? 

Reviewer #1: Yes

Reviewer #2: Yes

3. Have the authors made all data underlying the findings in their manuscript fully available?

Reviewer #1: No

Reviewer #2: Yes

4. Is the manuscript presented in an intelligible fashion and written in standard English?

Reviewer #1: Yes

Reviewer #2: Yes

5. Review Comments to the Author

Reviewer #1: The PLOS ONE manuscript entitled "Virus transmission via honey bee prey and potential impact on cocoon-building in labyrinth spiders Agelena Labyrinthica” describes work that is in generally well organized and clearly described. However, several concerns should be addressed.

1) Line 193: “In 62.5% of the tested spiders at least one virus was detected. DWV-B was the most common with more than half of the samples being positive (56.25%), followed by ABPV (25%), DWV-A (25%) and SBV (18.75%). In 40% of the spiders, presence of more than one virus was found.” This text and accompanying table (Table 2) requires further contextualization. As presented in the table the sample number is n=5. But the percentages are calculated in the text is on a per individual basis (n=16). It would be helpful if the rationale for pooling (as shown in Table 2) be given and that the table needs expanding. Are these pooled because they came from the same location? If so, please provide location names and if possible GPS co-ordinates. Also, it would be helpful if the location of the n=24 samples used in the feeding experiment be given as well. Based on this information, questions that come to mind is whether the feeding experiment spiders came from known virus positive locations? And what are the profile of virus infections in honey bee apiaries, near these spiders, at the time of sampling?

2) Line 250: “Unsurprisingly, DWV-B was detected the most, matching the prevalence of this virus in honey bees [47].” This is not clear given the resolution of both studies. Ref 47 is a review. This sentence needs rephrasing based on the regional or even better apiary data for virus prevalence relative to the location of sampling.

3) Sequence data (raw reads and in particularly DWV sequences, given that 100% match was not found) was not submitted to a public database. The M&M section does not describe the sequencing technique used. Was the PCR products cloned before sequencing? If so, how many clones were sequenced?

Reviewer #2: This is an interesting and useful paper, which I’d recommend be published after minor revision. I hope the comments below are useful.

Lines, 21-34. The abstract nicely describes the study goals and main results. Could any additional information be included on the frequency of infections from field collected samples? I think readers would like to know if infections are a rare or common occurrence. The abstract also leads off with “host-shifts of RNA viruses…”. Are these really “host-shifts”? Do we really know what the original host for all these viruses is? They were initially described from bees, but that doesn’t mean bees were the original host or that these spiders haven’t always been infected. Calling them “honey bee viruses” as you do in the introduction could be immensely misleading and incorrect.

Introduction, Lines 75-76. The introduction nicely sets the context for the manuscript. Further to my comments on the abstract, calling the viruses “honey bee viruses”, and pictures like that In Figure 1 (though a nice picture that should be included), makes the reader think that spiders in the field study have likely acquired viral infections from eating honey bees. That’s not necessarily the case, because due to the wide host range of many of these viruses the spenders could have picked them up from a wide range of arthropods.

Methods, Lines 101-107. The spiders were feed fruit flies and circlets. Both can host insect viruses, as the authors cite in reference 49. Is that a problem? There were only 16 spiders in the samples used for viral surveys. Is that enough to be confident of the absence of four of the seven viruses examined in these spiders? I think it probably is an insufficient sample size for any major conclusions on viral absence.

Methods. The procedures, approaches and statistical analyses seemed fine. The only things that would have been good to see would have been positive controls for the viral assays, and some sort of viral replication assay. Do the viruses replicate in the spiders? Are the actual spiders hosts that are parasitized by the viruses? I see later that the authors discuss this issue in the discussion. The sample sizes aren’t high— especially when it comes to results on cocoon building— but are probably enough to make the conclusions that the authors describe.

Results, Lines 197-199. Table 2 lists the viruses in alphabetical order. Perhaps it would be better to instead list by some viral taxonomic classification or affiliation?

Results, Lines 200-213. It is a pity that spiders free of these viruses could not be sourced. How many crickets were examined for these viruses?

Discussion, Lines 254-269. The authors tackle the issue of viral replication in this paragraph. I understand their argument regarding the potential for predators to ingest negative strands in the prey diet. But whether or not replication occurs here is highly relevant. How would the authors overcome the issues they describe? Perhaps measuring the amount of negative strand in the prey co,pared to the host? How long would a generation or two of the virus live for? If you wait a week after the spider has eaten, there shouldn’t be any negative strand in the spider from the food and only from replicating viruses within spider cells, right?

Conclusions, Lines 301-308. I don’t think that the conclusion section really adds much. It just repeats key points from the discussion. As referred to above, somewhere in the discussion I’d like to see some mention that these may not actually be “honey bee” viruses. Perhaps they are broad arthropod viruses that just happen to have first been found in honey bees.

References, Lines 312-432. The references include many of the key studies and are nicely cited. I’d encourage the authors to work through them to standardize the capitalization in many of the citations. There is the occasional review that is missing and which could add value, such as Nanetti et al. 2021. Pathogens Spillover from Honey Bees to Other Arthropods. Pathogens 10: 1044. https://doi.org/10.3390/pathogens10081044.

6. PLOS authors have the option to publish the peer review history of their article (what does this mean?). If published, this will include your full peer review and any attached files.

Reviewer #1: No

Reviewer #2: No

---

## [Author Response · Author response to Decision Letter 0]

3 Feb 2023

Please find our point to point response to the comments of the editors and reviewers below and as a uploaded word file (PONE-D-22-27101 response to Reviewers)

PONE-D-22-27101

Virus transmission via honey bee prey and potential impact on cocoon-building in labyrinth spiders (Agelena Labyrinthica)

PLOS ONE

 

Response to comments of Editorial Board Members

Thank you for submitting your manuscript to PLOS ONE. After careful consideration, we feel that it has merit but does not fully meet PLOS ONE’s publication criteria as it currently stands. Therefore, we invite you to submit a revised version of the manuscript that addresses the points raised during the review process. Please submit your revised manuscript by Jan 19 2023 11:59PM. If you will need more time than this to complete your revisions, please reply to this message or contact the journal office at plosone@plos.org. Response: Thank you for the kind invitation to resubmit the manuscript. We have now addressed all points raised during the reviewing process (see below) and would like to thank the editors and reviewers for their constructive contributions.

Response: All files got uploaded as requested. 

Response: There were no changes to the financial disclosure. The figure files got uploaded to the Preflight Analysis and Conversion Engine (PACE) digital diagnostic tool

Response: Not applicable as there were no new laboratory protocols developed. 

We look forward to receiving your revised manuscript.

Kind regards,

Olav Rueppell

Academic Editor

PLOS ONE

Journal Requirements: When submitting your revision, we need you to address these additional requirements.

Response: We used the style templates to compile our manuscript and named the files as requested above: 'Response to Reviewers', 'Revised Manuscript with Track Changes' and 'Manuscript'.

Response: We have added the following statement to the methods section (line 115*) ‘No ethical approval or permit was required to sample and work with this invertebrate species and all experiments were performed in accordance with relevant guidelines and regulations’.

*Line numbers are based on the simple- or no-markup version of the new manuscript version

Response: The data are now publicly available and can be accessed using the following informations: 10.6084/m9.figshare.21252489. The data availability statement has been changed accordingly (line 460) and the changes were mentioned in the cover letter.

4. Please remove your figures from within your manuscript file, leaving only the individual TIFF/EPS image files, uploaded separately. These will be automatically included in the reviewers’ PDF.

Response: The figures were deleted from the manuscript file and uploaded separately. The figure captions were moved to the end of the manuscript.

Additional Editor Comments:

Particularly reviewer #1 raised some important points, including that we cannot publish the study at all unless the raw data are publicly accessible.

Response: As mentioned above, the raw data are now publicly accessible and we have addressed all the comments of the reviewers (see below).

 

Reviewers' comments: Reviewer's Responses to Questions

Comments to the Author

1. Is the manuscript technically sound, and do the data support the conclusions?

Reviewer #1: Yes

Reviewer #2: Yes

2. Has the statistical analysis been performed appropriately and rigorously?

Reviewer #1: Yes

Reviewer #2: Yes

3. Have the authors made all data underlying the findings in their manuscript fully available?

Reviewer #1: No

Reviewer #2: Yes

4. Is the manuscript presented in an intelligible fashion and written in standard English?

Reviewer #1: Yes

Reviewer #2: Yes

Response: Thank you for the feedback. We are glad to hear that the only issue raised was that the data was not yet publicly accessible. This issue has now been addressed (see above). 

 

5. Review Comments to the Author

Response to comments of Reviewer 1

Reviewer #1: The PLOS ONE manuscript entitled "Virus transmission via honey bee prey and potential impact on cocoon-building in labyrinth spiders Agelena Labyrinthica” describes work that is in generally well organized and clearly described. However, several concerns should be addressed.

Response: We are delighted about this positive comment and sincerely grateful for the reviewers valuable inputs.

1) Line 193: “In 62.5% of the tested spiders at least one virus was detected. DWV-B was the most common with more than half of the samples being positive (56.25%), followed by ABPV (25%), DWV-A (25%) and SBV (18.75%). In 40% of the spiders, presence of more than one virus was found.” This text and accompanying table (Table 2) requires further contextualization. As presented in the table the sample number is n=5. But the percentages are calculated in the text is on a per individual basis (n=16). It would be helpful if the rationale for pooling (as shown in Table 2) be given and that the table needs expanding. Are these pooled because they came from the same location?

Response: Indeed, the reviewer is right that the numbers in the text are based on individual basis. But so are the numbers in the table. Unfortunately, the table caption was not clear enough and therefore got misinterpreted by the reviewer as the number of samples that were pooled together. However, it was supposed to be number of individually analysed samples with the same viral profile. For example, there were 5 individual samples that were tested for DWV-B only. We have updated the table caption to clarify this for future readers.

If so, please provide location names and if possible GPS co-ordinates. Also, it would be helpful if the location of the n=24 samples used in the feeding experiment be given as well. 

Response: The GPS coordinates have now been provided in the Material and Methods section (line 95).

Based on this information, questions that come to mind is whether the feeding experiment spiders came from known virus positive locations? And what are the profile of virus infections in honey bee apiaries, near these spiders, at the time of sampling?

Response: These are indeed good questions. We totally agree that it would have been great mapping all viruses used in this study in our study area. Unfortunately, we do not have these data for the years 2016 & 2017. This limitation has now been clearly mentioned in the discussion (line 292).

2) Line 250: “Unsurprisingly, DWV-B was detected the most, matching the prevalence of this virus in honey bees [47].” This is not clear given the resolution of both studies. Ref 47 is a review. This sentence needs rephrasing based on the regional or even better apiary data for virus prevalence relative to the location of sampling.

Response: The reviewer is right that the provided reference does not provide exact data on the prevalence of DWV-B at the sampling locations. Nonetheless, especially given that we do not have the data, we argue that the provided references (Ref 28 and 47) are correct, as they confirm that DWV occurs across Europe (incl. Switzerland) and is consistently the most prevalent viral pathogen found in western honey bees, Apis mellifera. This is in line with the general wording we have used in this context, which is not specifically refering to our sampling location.

3) Sequence data (raw reads and in particularly DWV sequences, given that 100% match was not found) was not submitted to a public database. The M&M section does not describe the sequencing technique used. Was the PCR products cloned before sequencing? If so, how many clones were sequenced?

Response: This is a good point raised by the reviewer. The sequences have now been uploaded to GenBank and the respective accession numberes (ABPV - OQ272302; DWV-B - OQ272303) are included in the manuscript (line 177). Furthermore, we have added a brief section to the material and methods section to state how we prepared the samples for sequencing. 

Response to comments of Reviewer 2

Reviewer #2: This is an interesting and useful paper, which I’d recommend be published after minor revision. I hope the comments below are useful.

Resonse: Happy to hear this positive comment. Constructive comments are always welcome and helpful.

Lines, 21-34. The abstract nicely describes the study goals and main results. Could any additional information be included on the frequency of infections from field collected samples? I think readers would like to know if infections are a rare or common occurrence. The abstract also leads off with “host-shifts of RNA viruses…”. Are these really “host-shifts”? Do we really know what the original host for all these viruses is? They were initially described from bees, but that doesn’t mean bees were the original host or that these spiders haven’t always been infected. Calling them “honey bee viruses” as you do in the introduction could be immensely misleading and incorrect.

Resonse: A series of good points raised by the reviewer. We have included a statement about the virus prevalence for field collected samples in the abstract. Of course, the reviewer is right, it is not known if honey bees are the original host. To make this more clear, we have included a respective section in the discussion (line 254+) and we have replaced virus host-shift with interspecies virus transmission for all instances, where the change was applicable. Last but not least, despite our explanation why we were refering to the viruses as "honey bee viruses" in the original version and the frequent use of the term "honey bee viruses" in the literature (e.g. see ref 16 or 29), we have changed the wording for all instances of honey bee viruses, to make it clearer that we are actually speaking about viruses commonly associated with honey bees without infering that honeybees were the original host.

Introduction, Lines 75-76. The introduction nicely sets the context for the manuscript. Further to my comments on the abstract, calling the viruses “honey bee viruses”, and pictures like that In Figure 1 (though a nice picture that should be included), makes the reader think that spiders in the field study have likely acquired viral infections from eating honey bees. That’s not necessarily the case, because due to the wide host range of many of these viruses the spenders could have picked them up from a wide range of arthropods.

Resonse: The reviewer is right that we cannot say anything about the origin of the viruses detected in the samples from the field. Thus, we have now included a statement in the discussion to clarify that the spiders could have been infected via any other exposure route (line 288+).

Methods, Lines 101-107. The spiders were feed fruit flies and circlets. Both can host insect viruses, as the authors cite in reference 49. Is that a problem? There were only 16 spiders in the samples used for viral surveys. Is that enough to be confident of the absence of four of the seven viruses examined in these spiders? I think it probably is an insufficient sample size for any major conclusions on viral absence.

Resonse: Crickets were tested negative for ABPV and DWV (line 208) and the fruit flies were derived from a laboratory cultured maintained in isolation over multiple generations implying that it is very unlikely that either explain our results or pose any problem. Obviously, the reviewer is right that the absence of the viruses in our samples does not prove that spiders cannot carry these viruses. Therfore, we carefully avoid such a claim in the discussion, nor do we make any conclusion on virual absence.

Methods. The procedures, approaches and statistical analyses seemed fine. The only things that would have been good to see would have been positive controls for the viral assays, and some sort of viral replication assay. Do the viruses replicate in the spiders? Are the actual spiders hosts that are parasitized by the viruses? I see later that the authors discuss this issue in the discussion. The sample sizes aren’t high— especially when it comes to results on cocoon building— but are probably enough to make the conclusions that the authors describe.

Resonse: As the reviewer pointed out, we discuss the limitations of our studies, provide an explanation why no viral replication assay was conducted and we highlight that the sample size was small therby creating the need for further evidence to confirm viral replication and clinical symptoms.

Results, Lines 197-199. Table 2 lists the viruses in alphabetical order. Perhaps it would be better to instead list by some viral taxonomic classification or affiliation?

Response: The table now includes an additional column referring to the virus family if applicable.

Results, Lines 200-213. It is a pity that spiders free of these viruses could not be sourced. How many crickets were examined for these viruses?

Response: We screened 16 crickets. This information is now included in the manuscript. 

Discussion, Lines 254-269. The authors tackle the issue of viral replication in this paragraph. I understand their argument regarding the potential for predators to ingest negative strands in the prey diet. But whether or not replication occurs here is highly relevant. How would the authors overcome the issues they describe? Perhaps measuring the amount of negative strand in the prey compared to the host? How long would a generation or two of the virus live for? If you wait a week after the spider has eaten, there shouldn’t be any negative strand in the spider from the food and only from replicating viruses within spider cells, right?

Response: One way of overcoming this issue would be the use of fluorescence- in situ -hybridization as we state on line 309 or maybe an additional longterm study, where virus levels are monitored over multiple weeks. Unfortunately both options are not feasible within the scope of this study. The suggestion to use the amounts of negative strand in prey and host would not help to derive any meaningful conclusion. Just the tiniest bit of negative strand could indicate ongoing replication with no means of differentiation between actual replication in the spider or false positive due to previous replication of the virus in the honey bee prey. Also, the ratio of positive to negative strand is expected to be 1000:1 if there is ongoing replication, but we are not aware of any evidence suggesting that this should be any different for replication in different hosts, i.e. bees, spiders or bee leftovers in spider guts. Lastly, we do not know how long we can expect false positives to be detected due to recently consumed viral particles. Accordingly, we waited four days since the last feeding and even if we would have waited longer, the situation would not have changed. We therefore believe that high level infections are the most conclusive for our data and remain carefull not to use strong wording.

Conclusions, Lines 301-308. I don’t think that the conclusion section really adds much. It just repeats key points from the discussion. As referred to above, somewhere in the discussion I’d like to see some mention that these may not actually be “honey bee” viruses. Perhaps they are broad arthropod viruses that just happen to have first been found in honey bees.

Response: The reviewer is right that the conclusion is just a final takaway message and we have no strong feeling wheter to keep it in or not. We therefore would like to leave it up to the editors to decide whether to delete it or not. The issue about honey bee viruses has been adressed earlier (see above).

References, Lines 312-432. The references include many of the key studies and are nicely cited. I’d encourage the authors to work through them to standardize the capitalization in many of the citations. There is the occasional review that is missing and which could add value, such as Nanetti et al. 2021. Pathogens Spillover from Honey Bees to Other Arthropods. Pathogens 10: 1044. https://doi.org/10.3390/pathogens10081044.

Response: We carefully worked through the citations to correct the capitalization. In accordance with the Vancouver style we have changed all titles so that only the first word as well as proper names are capitalized, we have removed any italics (including species names) as shown in the PLOS ONE guidelines, and we have replaced the full journal names with their corresponging abbreviations. Finally, we have included the review suggested by the reviewer as it is indeed of interest to our study.

Final statement: We do sincerely hope that we were able to address fully all of the reviewers' concerns. We would like to thank the colleagues for the time they have invested in our manuscript to provide detailed and constructive comments.

---

## [Editor Report · Decision Letter 1]

14 Feb 2023

Virus transmission via honey bee prey and potential impact on cocoon-building in labyrinth spiders (*Agelena Labyrinthica*)

PONE-D-22-27101R1

Dear Dr. Schlaeppi,

We’re pleased to inform you that your manuscript has been judged scientifically suitable for publication and will be formally accepted for publication once it meets all outstanding technical requirements. I think it is fine to leave in the conclusion section.

Kind regards,

Olav Rueppell

Academic Editor

PLOS ONE
---

## [Editor Report · Acceptance letter]

20 Feb 2023

PONE-D-22-27101R1 

Virus transmission via honey bee prey and potential impact on cocoon-building in labyrinth spiders (*Agelena Labyrinthica*) 

Dear Dr. Schläppi:

I'm pleased to inform you that your manuscript has been deemed suitable for publication in PLOS ONE. Congratulations! Your manuscript is now with our production department. 

Kind regards, 

on behalf of

Dr. Olav Rueppell 

Academic Editor

PLOS ONE